# Alpha-Ketoglutarate or 5-HMF: Single Compounds Effectively Eliminate Leukemia Cells via Caspase-3 Apoptosis and Antioxidative Pathways

**DOI:** 10.3390/ijms23169034

**Published:** 2022-08-12

**Authors:** Joachim Greilberger, Ralf Herwig, Mehtap Kacar, Naime Brajshori, Georg Feigl, Philipp Stiegler, Reinhold Wintersteiger

**Affiliations:** 1Division of Medical Chemistry, Otto Loewi Research, Medical University of Graz, 8010 Graz, Austria; 2Institute of Scientific Laboratory Dr. Greilberger, Schwarzl Medical Center, 8053 Lassnitzhoehe, Austria; 3Laboratories PD Dr. R. Herwig, 80337 Munich, Germany; 4Heimerer-College, 10000 Pristina, Kosovo; 5Department of Physiology, Faculty of Medicine, Yeditepe University, Ataşehir, 34755 İstanbul, Turkey; 6Department of Pathophysiology, Health Sciences Institute, Yeditepe University, Ataşehir, 34755 İstanbul, Turkey; 7Institute of Anatomy and Clinical Morphology, University of Witten/Herdecke, 58455 Witten, Germany; 8Division of Transplantation Surgery, Medical University of Graz, 8010 Graz, Austria; 9Department of Pharmaceutical Chemistry, Institute of Pharmaceutical Sciences, University of Graz, 8010 Graz, Austria

**Keywords:** alpha-ketoglutarate (aKG), 5-hydroxy-methyl-furfural (5-HMF), reactive oxygen and nitrogen species (RONS), peroxynitrite (ONOO^−^), nitrated tyrosine, leukemia, proliferation, caspase activity

## Abstract

Background: We recently showed that a combined solution containing alpha-ketoglutarate (aKG) and 5-hydroxymethyl-furfural (5-HMF) has a solid antitumoral effect on the Jurkat cell line due to the fact of its antioxidative, caspase-3 and apoptosis activities, but no negative effect on human fibroblasts was obtained. The question arises how the single compounds, aKG and 5-HMF, affect peroxynitrite (ONOO^−^) and nitration of tyrosine residues, Jurkat cell proliferation and caspase-activated apoptosis. Methods: The ONOO^−^ luminol-induced chemiluminescence reaction was used to measure the ONOO^−^ scavenging function of aKG or 5-HMF, and their protection against nitration of tyrosine residues on bovine serum albumin was estimated with the ELISA technique. The Jurkat cell line was cultivated in the absence or presence of aKG or 5-HMF solutions between 0 and 3.5 µM aKG or 0 and 4 µM 5-HMF. Jurkat cells were tested for cell proliferation, mitochondrial activity and caspase-activated apoptosis. Results: aKG showed a concentration-dependent reduction in ONOO^−^, resulting in a 90% elimination of ONOO^−^ using 200 mM aKG. In addition, 20 and 200 mM 5-HMF were able to reduce ONOO^−^ only by 20%, while lower concentrations of 5-HMF remained stable in the presence of ONOO^−^. Nitration of tyrosine residues was inhibited 4 fold more effectively with 5-HMF compared to aKG measuring the IC50%. Both substances, aKG and 5-HMF, were shown to cause a reduction in Jurkat cell growth that was dependent on the dose and incubation time. The aKG effectively reduced Jurkat cell growth down to 50% after 48 and 72 h of incubation using the highest concentration of 3.5 µM, and 1, 1.6, 2, 3 and 4 µM 5-HMF inhibited any cell growth within (i) 24 h; 1.6, 2, 3 and 4 µM 5-HMF within 48 h (ii); 2, 3 and 4 µM 5-HMF within 72 h (iii). Furthermore, 4 µM was able to eliminate the starting cell number of 20,000 cells after 48 and 72 h down to 11,233 cells. The mitochondrial activity measurements supported the data on aKG or 5-HMF regarding cell growth in Jurkat cells, in both a dose- and incubation-time-dependent manner: the highest concentration of 3.5 µM aKG reduced the mitochondrial activity over 24 h (67.7%), 48 h (57.9%) and 72 h (46.8%) of incubation with Jurkat cells compared to the control incubation without aKG (100%). 5-HMF was more effective compared to aKG; the mitochondrial activity in the presence of 4 µM 5-HMF decreased after 24 h down to 68.4%, after 48 h to 42.9% and after 72 h to 32.0%. Moreover, 1.7 and 3.4 µM aKG had no effect on caspase-3-activated apoptosis (0.58% and 0.56%) in the Jurkat cell line. However, 2 and 4 µM 5-HMF increased the caspase-3-activated apoptosis up to 22.1% and 42.5% compared to the control (2.9%). A combined solution of 1.7 µM aKG + 0.7 µM 5-HMF showed a higher caspase-3-activated apoptosis (15.7%) compared to 1.7 µM aKG or 2 µM 5-HMF alone. In addition, 3.5 µM µg/mL aKG + 1.7 µM 5-HMF induced caspase-activated apoptosis up to 55.6% compared to 4.5% or 35.6% caspase-3 activity using 3.5 µM aKG or 4 µM 5-HMF. Conclusion: Both substances showed high antioxidative potential in eliminating either peroxynitrite or nitration of tyrosine residues, which results in a better inhibition of cell growth and mitochondrial activity of 5-HMF compared to aKG. However, caspase-3-activated apoptosis measurements revealed that the combination of both substances synergistically is the most effective compared to single compounds.

## 1. Introduction

Alpha-ketoglutarate (aKG) is commonly known for its function in the Krebs cycle and antioxidative capacity. Furthermore, the oxoglutarate dehydrogenase complex (OGDC) or α-ketoglutarate dehydrogenase complex is an enzyme complex that plays a major role in the citric acid cycle. aKG is a potent scavenger in eliminating peroxynitrite [1], a physiological nitrating agent derived from nitrosyl radial (NO^•^) and superoxide anion radical (O_2_*^−^; [2]), to succinate and nitrite, which can be easily converted back to NO^•^.

5-Hydroxymethylfurfural (HMF) is formed from reducing sugars in acidic environments when heated through the Maillard reaction. HMF has shown antioxidative, anti-allergic, anti-inflammatory, antihypoxic, antisickling and antihyperuricemic effects. In addition, 5-HMF is known as a radical scavenger against hydroxyl radicals, alkyl radicals and superoxide anion radicals but also to inhibit oxidant enzymes, such as myeloperoxidase, and to increase the expression of glutathione and superoxide dismutase (SOD) in cell cultures [3].

We have recently shown that addition of a combined solution containing aKG + 5-HMF to leukemic cells exerted antitumoral, antiproliferative and apoptotic effects compared to HF-SAR [4]. This seems to be associated with the antioxidative capacity of both substances, including the prevention of oxidative protein modifications, but also protein nitration and oxidative DNA damage [5]. This is closely connected with increased nitrotyrosine levels, release of cytochrome c, activation of caspase-3 and alteration of the p53 gene.

During carcinogenesis, RONS, such as O_2_*^−^, are produced in a higher number, which are speculated to be involved in all steps including the initiation, promotion and activation of proto-oncogenes and tumor suppressor genes [6].

A newly presented meta-analysis correlated antioxidative activating enzymes, such as SOD, CAT, GPx and levels of RONS markers, namely, malondialdehyde (MDA) and 8-hydroxy-desoxoguanosine (8-OHdG), between cancer patients and a healthy population [7]. Overall, MDA, a lipid peroxidation end product, and 8-OHdG, a nucleic acid oxidation marker, were markedly higher in a cancer group that also included CLL patients, whereas all antioxidative enzymes levels were lower compared to the healthy population. We estimated that the Jurkat cell line expresses higher carbonylated proteins in the cell membrane compared to normal cells [4]. Malondialdehyde is among the most expressed subgroups of these carbonyl groups on membrane proteins. Thus, we suggest that antioxidative substances, such as alpha-ketoglutarate and 5-HMF, can counteract the cell growth of leukemic cells directly in the mitochondrial activity in terms of aKG and 5-HMF. Additionally, both substances detoxify ammonia on one the hand by the transfer ionically with aKG to the glutamate–glutamine–oxalacetate pathway (GOT) or, on the other hand, covalently to 5-HMF.

aKG is required as a potent oxidant for the reductive carboxylation bidirectional in cancer cells with mitochondrial defects to operate IDH inversely and SDH directly [8] Under intracellular acidic conditions, which does take place intracellular in cancer cells, the nonenzymatic decarboxylation of aKG in the presence of H_2_O_2_ and ONOO^−^ cannot occur. Thus, nonenzymatically generated succinate decreases, which itself is able to stabilize HIFα [1].

Both aKG and 5-HMF are also involved in increasing SOD, GPx and CAT to reduce RONS in cell culture and cancer patients [3,9]. Free and bound carbonyl proteins levels were measured in patients with cancer before and after receiving both substances: aKG and 5-HMF [9,10,11]. Supplemented or iv-applicated aKG + 5-HMF effectively reduced the carbonylated proteins in cancer patients compared to the placebo groups [10]. Carbonyl proteins were also effectively reduced by the combination of aKG and HMF in cell culture using aKG + 5-HMF together.

The question arises over which substance of the combined solution of aKG + 5-HMF has (i) antiproliferative, (ii) caspase-3 and apoptosis effects on leukemic cells or (iii) an antinitrating effect on scavenging peroxynitrite to prevent nitro-tyrosine, or if they act in a synergistic manner. In addition to the oxidizing ability of ONOO^−^, nitration by nitrating agents of biomolecules, such as ONOO^−^, is involved directly in cytochrome c release, activation of caspase-3 and p53 alterations.

## 2. Results

### 2.1. ONOO^−^ Luminol-Induced Chemiluminescence in the Presence or Absence of αKG, 5-HMF and a Combination of aKG + 5-HMF

Figure 1A shows ONOO^−^ luminol-induced chemiluminescence levels after 1 s in the presence or absence of different concentrations of αKG (*n* = 5). Using 4 mM αKG (383,632 ± 79,292 cps), a significant reduction in ONOO^−^ was obtained compared to the control signal in the absence of any antioxidant (435,557 ± 71,480 cps; *p* < 0.01). Increasing the concentration to 20 mM αKG showed a more than two-thirds reduction (158,688 ± 46,197 cps; *p* < 0.01). Usage of 200 mM αKG reduced the signal completely (1267 ± 1929 cps; *p* < 0.01).

Figure 1B shows the measurement of ONOO^−^ luminol-induced chemiluminescence signals in the presence or absence of different concentrations of 5-HMF (*n* = 5). A significant reduction in ONOO^−^ was obtained using 20 mM and 200 mM 5-HMF (374,226 ± 57,247 cps and 344,820 ± 82,090 cps) compared to the control signal in the absence of any antioxidant (471,650 ± 29,094 cps; *p* < 0.01).

Figure 1C shows the combination of 5-HMF + aKG solutions and its effect on the reduction in peroxynitrite. The usage of 0.8 mM 5-HMF + 0.8 mM aKG (1,528,959 ± 142,670 cps) and 4 mM 5-HMF + 4 mM aKG (1,410,431 ± 120,155 cps) showed no significant reduction in peroxynitrite compared to the control (0 mM 5-HMF + 0 mM aKG: 1,467,988 ± 191,110 cps). Increasing the concentration to of 20 mM 5-HMF + 20 mM αKG showed a significant reduction by 25% (890,150 ± 207,464 cps; *p* < 0.01). Using 200 mM 5-HMF + 200 mM αKG reduced the signal significantly down to 8% (118,717 ± 28,395 cps; *p* < 0.01) compared to control and 11% (*p* < 0.01) compared to 20 mM 5-HMF + 20 mM αKG, respectively.

### 2.2. Nitrated Tyrosine Estimation on BSA in the Presence and Absence of αKG, 5-HMF and a Combination of aKG + 5-HMF

Figure 2B shows the absorbance signals of nitro-tyrosine in the absence or presence of 5-HMF. Using 3 mM 5-HMF, the nitro-tyrosine level decreased significantly down to 82% compared to 0.75 mM 5-HMF (0.791 ± 0.11 vs. 1.014 +/− 0.06; *p* < 0.01); 6 mM 5-HMF down to 15% (0.143 ± 0.035, *p* < 0.01) and 12 mM 5-HMF to 14% (0.139 ± 0.043; *p* < 0.01) as shown in Figure 2B. Both, 6 mM and 12 mM also showed a significant reduction compared to 1.5 mM 5-HMF (*p* < 0.01).

Combined equimolar aKG + 5-HMF solutions and its inhibiting effect in the nitration of tyrosine residues are presented in Figure 2B: using 3, 6 and 12 mM 5-HMF, nitro-tyrosine levels were reduced significantly down to 64% (0.552 ± 0.043; *p* < 0.01), 25% (0.214 ± 0.035; *p* < 0.01) and 4% (0.033 ± 0.063; *p* < 0.01). Furthermore, significance was found between 6 and 12 mM (*p* < 0.01). Measurements of nitrated tyrosine in the presence of several aKG concentrations were recently presented elsewhere [4] describing a significant but lower antinitrating effect using (0.61 ± 0.017), 24 (0.137 ± 0.014) and 36 mM aKG (0.162 ± 0.029) compared to equimolar 5-HMF concentrations.

### 2.3. Cell Proliferation Experiments

Figure 3A describes the cell growth with different concentrations of aKG in Jurkat cells over 3 days. After 24 h, no influence effecting on the cell growth, positively or negatively, was obtained using several concentrations (i.e., 0.9, 1.4, 1.7, 2.6 and 3.5 µM) of aKG. The cell growth after 48 h was significantly reduced compared to the control (57,123 ± 2122 cells, *n* = 5) by several concentrations of aKG: 1.4 µM aKG (53,644 ± 3234 cells; *p* < 0.01; *n* = 5), 1.7 µM aKG (43,145 ± 2550 cells; *p* < 0.01; *n* = 5), 2.6 µM aKG (41,233 ± 1811 cells; *p* < 0.01; *n* = 5), and 3.5 µM (34,246 +/− 1879 cells; *p* < 0.01; *n* = 5). The latter one showed nearly a 40% reduction and obtained a significant reduction compared to the usage of 2.6 µM aKG (*p* < 0.05; *n* = 5). After 72 h of incubation, all used aKG concentrations significantly decreased the growth of Jurkat cells; a 40% reduction was obtained using 3.5 µM aKG (45,023 ± 5497 cells; *p* < 0.01; *n* = 5), followed by 35% with 2.6 µM (49,323 ± 4237 cells; *p* < 0.01; *n* = 5), 23% with 1.7 µM aKG (58,098 ± 5127 cells; *p* < 0.01; *n* = 5), 19% with 1.4 µM (60,923 ± 3577 cells; *p* < 0.01; *n* = 5) and 12% using 0.9 µM aKG (66,256 ± 4239 cells; *p* < 0.01; *n* = 5) compared to the growth of control cells (75,346 ± 3786 cells; *n* = 5). While in the absence of aKG, cell growth increased 2.9 fold after 48 h and 3.8 fold after 72 h, the highest concentrations of aKG, 2.6 and 3.5 µM, depleted the cell growth compared to the control: 35% and 41% reductions were observed after 48 h, and 34% and 37% reductions were observed after 72 h of incubation.

Figure 3B describes the cell growth with different concentrations of 5-HMF in Jurkat cells over 3 days. The lowest used concentration of 1 µM 5-HMF significantly decreased cell growth compared to the control incubation after 48 h (28,235 ± 5587 cells vs. 421,365 +/− 8177 cells; *p* < 0.05; *n* = 5), more significance was obtained using 1.6 (18,678 ± 5578 cells; *p* < 0.01; *n* = 5), 2 (18,734 ± 5865 cells; *p* < 0.01; *n* = 5) and 3 µM (17,132 ± 4223 cells; *p* < 0.01; *n* = 5). The highest concentration (i.e., 4 µM; 11,765 ± 1545 cells; *p* < 0.05; *n* = 5) showed a significant decrease in cell growth compared to 3 µM 5-HMF after 48 h of incubation. We estimated a decrease in cell growth using 1 µM 5-HMF by nearly 33% (54,644 +/− 13,211 cells; *n* = 5), but it was not significant compared to 72 h of the control cell growth (82,113 +/− 15,188 cells; *n* = 5). Massive significant cell growth reduction was obtained dose dependently: 1.6 µM 5-HMF reduced cell growth down to 36% (29,222 +/− 8224 cells; *p* < 0.001; *n* = 5), 2 µM 5-HMF to 27% (22,215 +/− 5587 cells; *p* < 0.01; *n* = 5), 3 µM 5-HMF to 16,105 +/− 2100 cells; *p* < 0.01; *n* = 5) and 4 µM 5-HMF to 11,233 +/− 5545 cells; *p* < 0.01; *n* = 5). Both 3 and 4 µM 5-HMF showed a significant reduction in cell growth compared to 2 µM 5-HMF (*p* < 0.01).

After 24 h, no difference in the cell growth of Jurkat cells was obtained in the presence or absence of 5-HMF compared to the starting conditions. After 48 h of incubation, cell growth in the absence of 5-HMF increased up to 2.1 fold (421,245 ± 8122 cells) compared to starting conditions (20,000 cells). The usage of 1.6, 2, and 3 µM 5-HMF inhibited any cell growth of Jurkat cells after 48 h compared to the control, and 4 µM 5-HMF reduced cell growth after 48 h of incubation by 40%. After 72 h of incubation, no cell growth was estimated using 2 µM compared to the starting conditions, whereas 0 µg/mL 5-HMF increased 4.1 fold, 1 µM 5-HMF increased 2.7 fold and 1.6 µM 5-HMF increased 1.5 fold. The highest used 5-HMF concentrations, 3 and 4 µM, were able to reduce the cell growth of Jurkat cells by 33 and 44%, respectively.

### 2.4. Cytotoxic Assay

The mitochondrial activity of the Jurkat cells, as expressed in the absorbance at 450 nm in the absence or presence of several aKG dilutions, is presented in Figure 4A. After 24 h of incubation, 1.4 (0.191 +/− 0.012), 1.7 (0.172 +/− 0.015), 2.6 (0.168 +/− 0.08) and 3.5 µM (0.151 +/− 0.01) aKG showed a significant reduction in the mitochondrial activity compared to the control after 24 h (0.22 +/− 0.02; *n* = 5; *p* < 0.01). The significance was also estimated between 2.6 and 3.5 µM aKG (*p* < 0.01).

After 48 h, several used concentrations of aKG significantly reduced the mitochondrial activity in a concentration-dependent manner: 0.9 µM reduced the signal to 0.263 ± 0.020 (*p* < 0.01; *n* = 5), followed by 1.4 µM (0.245 ± 0.03; *p* < 0.01; *n* = 5), by 1.7 µM (0.230 ± 0.019; *p* < 0.01; *n* = 5), by 2.6 µM (0.210 ± 0.013; *p* < 0.01; *n* = 5) and 3.5 µM (0.180 ± 0.018; *p* < 0.01; *n* = 5) compared to the control (0.311 ± 0.023; *n* = 5). Additionally, 2.6 µM aKG presented a significantly lower absorbance compared to 0.9 µM aKG (*p* < 0.01) and a significantly higher absorbance compared to 3.5 µM aKG (*p* < 0.01).

Nearly the same situation was also estimated after 72 h of incubation: 3.5 µM aKG (0.185 ± 0.010; *p* < 0.01; *n* = 5) showed the significantly lowest absorbance compared to the control (0.395 ± 0.030, *n* = 5) but also compared to 2.6 µM aKG (0.244 ± 0.012; *p* < 0.01; *n* = 5), which itself was significantly lower compared to 1.7 µM (0.289 ± 0.021; *p* < 0.01; *n* = 5). Compared to the control, 0.9 µM (0.330 ± 0.021; *p* < 0.01), 1.4 (0.300 ± 0.030; *p* < 0.01; *n* = 5) and 1.7 µM aKG (0.289 ± 0.021; *p* < 0.01; *n* = 5) showed significantly lower absorbance of the mitochondrial activity.

The mitochondrial activity of the Jurkat cells, as expressed in the absorbance at 450 nm in the absence or presence of several 5-HMF dilutions, is presented in Figure 4B:

After 24 h of incubation, 1.6 (0.184 ± 0.015; *n* = 5), 2 (0.172 ± 0.030; *n* = 5), 3 (0.164 ± 0.018; *n* = 5) and 4 µM 5-HMF (0.160 ± 0.015; *n* = 5) showed a significant reduction in mitochondrial activity compared to the control after 24 h (0.22 +/− 0.02; *n* = 5; *p* < 0.01). The significance was also estimated between 3 and 4 µM aKG (*p* < 0.01) and between 1 µg/mL and 3 µM 5-HMF (*p* < 0.05).

After 48 h, several used concentrations of 5-HMF significantly reduced the mitochondrial activity in a concentration-dependent manner: 1 µM reduced the signal to 0.269 ± 0.023 (*p* < 0.01; *n* = 5), followed by 1.6 µM (0.217 ± 0.02; *p* < 0.01; *n* = 5), by 2 µM (0.193 ± 0.03; *p* < 0.01; *n* = 5), by 3 µM (0.173 ± 0.018; *p* < 0.01; *n* = 5) and 4 µM (0.148 ± 0.02; *p* < 0.01; *n* = 5) compared to the control (0.345 ± 0.015; *n* = 5). Additionally, 1.6 µM 5-HMF presented a significantly lower absorbance compared to 1 µM 5-HMF (*p* < 0.01), and a significantly higher absorbance was obtained with 25 µM 5-HMF compared to 4 µM 5-HMF (*p* < 0.01).

Nearly the same situation was also estimated after 72 h of incubation: 4 µM 5-HMF (0.164 ± 0.010; *p* < 0.01; *n* = 5) showed the significantly lowest absorbance compared to the control (0.512 ± 0.021, *n* = 5) but also compared to 3 µM 5-HMF (0.220 ± 0.018; *p* < 0.01; *n* = 5), which itself was significantly lower compared to 2 µM (0.292 ± 0.020; *p* < 0.01; *n* = 5). Compared to the control, 1 (0.353 +/− 0.019; *p* < 0.01), 1.6 (0.319 +/− 0.030; *p* < 0.01; *n* = 5) and 2 µM (0.292 +/− 0.020; *p* < 0.01; *n* = 5) 5-HMF showed significantly lower absorbance of mitochondrial activity. Between 1.6 and 2 µM 5-HMF there was also a significantly lower absorbance obtained (*p* < 0.05).

Figure 5A was calculated from all of the absorbance data as shown in Figure 4A, setting 100% for 0 µM aKG for each time point (i.e., 24, 48 and 72 h) of incubation on the Jurkat cell line. The same was conducted with 5-HMF (Figure 5B).

In the absence of aKG, all calculated percentages of 24, 48 and 72 h incubated Jurkat cell lines showed no significant difference (Figure 5A).

Using 2.6 µM aKG, the cell growth was significantly, by nearly 75.3 ± 4.8% after 24 h of incubation, lower compared to the 24 h control (100 ± 9.0%; *p* < 0.01; *n* = 5); 3.5 µM aKG reduced the cell growth significantly to 67.7 ± 6.6% (*p* < 0.01; *n* = 5) compared to 0 and 0.9 µM aKG (89.7 ± 6.0%; *p* < 0.01; *n* = 5). After 48 h, 0.9 µM aKG was able to reduce significantly the cell growth (84.6 ± 7.6; *p* < 0.05; *n* = 5) compared to the control for 48 h in the absence of aKG (100 ± 7.4%) followed by 1.7 (74.0 ± 8.0; *p* < 0.01; *n* = 5), 2.6 (67.5 ± 6.0%; *p* < 0.01; *n* = 5) and 3.5 µM aKG (57.9 ± 10.0%; *p* < 0.01; *n* = 5). Furthermore, 2.6 µM aKG (*p* < 0.05) showed a lower significance than 3.5 µM aKG (*p* < 0.01) compared to 0.9 µM aKG. After 72 h of incubation, all used aKG concentrations significantly reduced cell growth compared to the control (100 ± 7.6%; *p* < 0.01; *n* = 5): 83.5 ± 6.4% for 0.9 µM, 75.9 ± 10.0% for 1.4 µM, 73.2 ± 7.3% for 1.7 µM, 61.8 ± 4.9% for 2.6 and 3.5 µM aKG more than half (46.8 ± 5.4%). A significant difference was also calculated between 3.5 and 2.6 µM (*p* < 0.01) and between 2.6 and 0.9 µM aKG (*p* < 0.01).

In the absence of 5-HMF, all calculated percentages of 24, 48 and 72 h incubated Jurkat cell lines showed no significant difference (Figure 5B). Using 1.6 µg/mL 5-HMF, the cell growth of Jurkat cells was significantly, by nearly 78.6 ± 6.4%, lower after 24 h of incubation compared to the 24 h control (100 ± 8.6%; *p* < 0.01; *n* = 5), 2 µM 5-HMF down to 73.5 ± 12.8% (*p* < 0.01; *n* = 5), 3 µM 5-HMF to 70.1 ± 7.7% (*p* < 0.01; *n* = 5) and 4 µM 5-HMF reduced the cell growth to 68.4 ± 6.4% (*p* < 0.01; *n* = 5) compared to the 24 h control without 5-HMF. After 48 h, several used 5-HMF concentrations significantly reduced the cell growth of Jurkat cells (*p* < 0.01) in a concentration-dependent manner: 1 µM 5-HMF down to 78.0 ± 6.7% (*n* = 5), 1.6 µM 5-HMF to 62.3 ± 5.8%, 2 µM 5-HMF to 55.9% (*n* = 5), 3 µM 5-HMF to nearly half (50.1 ± 5.2%; *n* = 5) and 4 µM down to 42.9 +/− 5.8% (*n* = 5). A significant reduction was obtained also between 3 and 1 µM 5-HMF (*p* < 0.01; *n* = 5) and between 4 and 2 µM (*p* < 0.05; *n* = 5). After 72 h of incubation time, cell growth was significant lower using 1 µM 5-HMF compared to the control (68.9 ± 3.6% vs. 100 +/− 4.1%; *p* < 0.01; *n* = 5). This reduction continued using 1.6 µM (62.3 +/− 5.9%; *n* = 5), 2 µM (57.0 ± 3.9%; *n* = 5), 3 µM (43.0 ± 3.5%; *n* = 5) and by more than one-third with 4 µM 5-HMF (32.0 ± 2.0%, *n* = 5). The latter concentration was also significantly lower compared to 3 µM (*p* < 0.01), which itself was significantly lower compared to 2 µM 5-HMF (*p* < 0.01); 2 µM itself showed a significantly reduced cell growth compared to 1 and 1.6 µM 5-HMF (*p* < 0.01).

Comparing the mitochondrial activity with the cell growth of the Jurkat cells, Figure 6A shows a high linear relation (r^2^ = 0.9051) of all aKG concentrations at different incubation times, nearly the same for 5-HMF (r^2^ = 0.8518) as presented in Figure 6B. The slope in the linear function using aKG was even higher (k = 0.0047) compared to 5-HMF (k = 0.005). The 5-HMF clouds are shown more in the lower scales between 10 and 30 percent of cell growth, whereas aKG was randomly distributed between 30 and 70% of cell growth.

### 2.5. Caspase-3-Activated Apoptosis Measurements

The estimation of caspase-3-activated apoptosis (Figure 7A–C) significantly increased using 2 µM (10.3 ± 1.8%; *p* < 0.01; *n* = 3) and 4 µM 5-HMF (35.6 ± 3.2%; *p* < 0.01; *n* = 3) after 72 h of incubation of Jurkat cells compared to the control (2.9 ± 2.3%; *n* = 3; *p* < 0.01), whereas the usage of 1.7 and 3.5 µM aKG showed no significant difference (1.6 ± 0.9% and 4.5 +/− 0.8%). A combination of 1.7 µM aKG + 0.7 µM 5-HMF was significant higher (15.7 ± 2.8%; *p* < 0.01; *n* = 3) compared to the control but also compared to 2 µM 5-HMF (*p* < 0.01) or 1.7 µM aKG (*p* < 0.01). The highest caspase-3-activated apoptosis was measured using 3.5 µM aKG + 1.3 µM 5-HMF (55.6 ± 2.1%; *n* = 3), which was significantly higher compared to 4 µM 5-HMF (*p* < 0.01) or 3.5 µM aKG (*p* < 0.01) but also to 1.7 µM aKG + 0.7 µM 5-HMF (*p* < 0.01). The usage of 3.5 µM aKG + 1.3 µM 5-HMF showed a nearly 3.5-fold higher caspase-3-activated apoptosis compared to 1.7 µM aKG + 0.7 µM 5-HMF; 4 µM 5-HMF was 3.5-fold more effective in increasing the caspase-3-activated apoptosis compared to 2 µM 5-HMF.

Figure 7D shows the concentration-dependent caspase-activated apoptosis of aKG, 5-HMF and of the combined solution aKG + 5-HMF after 72 h of incubation. As the positive control CPT showed a logarithmic correlation perfectly with r^2^ = 1, saturation was reached using 4 µM. The usage of aKG showed no correlation, because no caspase-activated apoptosis was obtained. 5-HMF showed a linear correlation with a correlation coefficient of approximately r^2^ = 0.98. No saturation was obtained using 4 µM, and the caspase-activated apoptosis using 4 µM 5-HMF was only approximately 10% lower than the highest positive CPT signal. The combination solution (i.e., aKG + 5-HMF) showed also a linear function with a regression term of r^2^ = 0.98. The slope was higher compared to the 5-HMF alone. The signal using 3.5 µM aKG + 1.3 µM 5-HMF was 20% higher compared to 4 µM 5-HMF, the usage of 1.7 µM aKG + 0.7 µM 5-HMF was 5% higher than 2 µM 5-HMF. No saturation was obtained using 3.5 µM aKG + 1.3 µM 5-HMF compared to CPT.

## 3. Discussion

The use of aKG and 5-HMF as main components in an anticancerogenic solution for treating prostatic tumor patients by intravenous application was recently presented [12,13]. The doubling time of PSA increased significantly as the primary outcome. Using the same solution, a reduction in tumor mass in non-small cancer lung patients and also a higher quality of life using the Karnowski index was obtained in the verum group compared to the placebo group [13]. The combination of aKG and 5-HMF showed also a better prevention against cigarette smoke-induced radical protein modification compared to ascorbic acid or its single compounds [14]. aKG regulates as substrate processes in the inner mitochondrial membrane such as alpha-ketoglutarate dehydrogenase (AKGDH) or HIFα [15], in addition to oxygen to inhibit growth factor (e.g., endothelial growth factor) in general. Numerous 2-oxo-glutarate-dependent dioxygenases (2OGDDs) are recurrently dysregulated in malignancies, and aKG is also required for its function in 2OGDDs in neoplasia [16]. Furthermore, aKG seems to be involved in subduing tumors in bladder cancer patients [17]. In both enzymes, namely, HIF-α and 2OGDD, oxygen functions in addition to 2-oxoglutarate (αKG) act as a substrate to decarboxylase aKG into forming succinate and CO_2_ from O_2_ [18]. 2OGDD controls the hydroxylation on the hypoxia-inducible factor (HIF)-α subunit at the proline 402 or 564 of the prolyl hydroxylase domain (PHD), which plays an immense role in the regulation of hypoxia by promoting the von Hippel–Lindau protein, ubiquination and proteasomal degradation.

Nitrating agents, such as peroxynitrite, result in one way: by the excessive production of superoxide anion radicals (O_2_^•−^) and nitrosyl radicals (NO^•^). The nitration of organic compounds, such as (i) amino acids (e.g., tryptophane), (ii) carbohydrates, (iii) nucleic acids and (iv) fatty acids, is involved in normal metabolic pathways but also in the beginning or generation of several pathologies such as cardiovascular disease, hypo- and hypertension, neurodegenerative processes, such Morbus Parkinson’s or Morbus Alzheimer’s disease, inflammation, cancer-causing pathologies and septicemias [19]. Arginine, the main compound in the enzymatic production of NO^•^ via the NOS pathway (NO-synthase), itself is able to be nitrated to nitro-arginine, which is a potent inhibitor for the endothelial, neuronal, macrophage and inducible NOS.

Moreover, it results in the oxidation of thiols peroxynitrite nitrate and oxidize tyrosine residues. The estimation of nitro tyrosine residues on several proteins is evidence of extensively generated peroxynitrite in vivo and in vitro [20,21]. We recently showed that aKG is able to recycle nonenzymatically NO^•^ from peroxynitrite only under conditions over pH 7 in a concentration-dependent manner and to prevent nitration of tyrosine residues on mitochondrial and cytosolic proteins [1]. At lower pH, aKG was not able to interact with ONOO^−^. Here, we presented, for the first time, that 5-HMF is not as good at reducing peroxynitrite directly compared to aKG, but nitration of tyrosine to nitro-tyrosine was better blocked by 5-HMF compared to aKG. Combining both substances may implicate these different interactions with ONOO^−^ in a more effective way. However, at low concentrations (0–20 mM aKG + 5-HMF), no synergistic effect was obtained in the elimination of ONOO^−^. Even at higher dose (i.e., 200 mM aKG + 200 mM 5-HMF), a better synergistic effect to eliminate peroxynitrite was estimated compared to single compounds. Using the estimation of nitro-tyrosine as a sensor for excessive peroxynitrite, 5-HMF was 4–5-fold more effective at inhibiting nitration compared to aKG. Even the combined substances showed a better reduction at the highest used concentration compared to the single compounds indicating again the different mechanism of aKG and 5-HMF. The same synergistic effect was also estimated using other radical induced measurements such as cigarette smoke radicals on proteins generating carbonyl proteins [4]. In addition to more than 4000 radicals in cigarette smoke, there is peroxynitrite, which can oxidize and/or nitrate lipids, proteins and nucleic acids. In the same report, we have shown that a combination of aKG and 5-HMF effectively decreased cell growth and mitochondrial activity of Jurkat cell lines by their reduction in oxidative stress. Here, we present measurements of the single components for the first time. 5-HMF was able to inhibit the concentration- and incubation-time-dependent cell growth: after 72 h of incubation using 500 µg/mL 5-HMF, cell growth was significantly lower compared to the starting cell counts. Usage of aKG also reduced cell growth, but in a slightly lower manner. It is known that aKG is involved in the regulation of oxygen levels via the HIF alpha pathway as a substrate in addition to oxygen, vitamin C and ferric ions. Therefore, we estimated with aKG alone a reduction in the mitochondrial activity of Jurkat cells lines which correlated highly positive with cell growth in a linear function: the higher the aKG concentration the lower the cell growth and mitochondrial activity. We also found a similar correlation with 5-HMF, but the slope was lower compared to aKG. It was recently shown that 5-HMF can stabilize HIF alpha [22], and therefore use other pathways to inhibit cell growth and cytotoxic activity. These pathways were characterized by [23], which increased enzyme activities of SOD, catalase (CAT) and glutathione peroxidase (GPx) to protect cells from oxidative damage and to induce high antiproliferative activity on cancer cells.

The aKG was shown to induce apoptosis via caspase-9 and JNK mechanism by 7% and 12.1% using 5 and 200 mM [4], and we also observed apoptosis effects, which were not significantly different compared to the control using 0–3.4 mM solutions (3.4 mM = 500 µg/mL). Caspase-3 activity was not increased by all used aKG concentrations, indicating that the JNK and caspase pathways JNK and caspase-9 were used. 5-HMF showed a massive caspase-3 activity of 22% and 45% using 250 µg/mL and 500 µg/mL concentration, inducing apoptosis of Jurkat cells up to 10% and 35%, but we obtained the highest caspase-3 activity and apoptosis using the combined solution 500 µg/mL aKG + 166.7 µg/mL of 51% and 55%.

NO^•^ is involved in human leukemia [24] by a different pathway, e.g., apoptotic or nonapoptotic. Inducible NOS (iNOS) is regulated via transcriptional pathways and calcium and calmodulin. Under normal physiological conditions iNOS is not present on cells. During inflammation the expression of iNOS increase in the presence of cytokines, generated by macrophages. Hypoxia itself is a main promotor for the induction of iNOS, which is directly switched with the HIF-alpha pathway. It is known that under hypoxic conditions, the generation of radicals increases. Radicals, such as NO^•^, regulates cytostatic pathways against human malignant and nonmalignant cells [25], which remains unclear, but it seems that this might be concentration dependent. It was recently shown [26] that Jurkat cells exposed to NO^•^ implicated a decrease in cardiolipin. Loss of cardiolipin massively affected the mitochondrial energy cycle process, which reduced the respiratory chain complex activities accompanied by a massive release of radicals, generated from dysregulated inner-mitochondrial enzymes such as AKGDH and/or low mitochondrial transmembrane potential. Activation of caspase-9 and caspase-3 follows after the release of cytochrome c into the cytosol. It seems that NO^•^ plays a crucial role in the activation of apoptotic processes in human leukemic cells and a loss of NO^•^ will help human leukemic cells to survive. A loss of NO can be speculated by the formation of ONOO^−^. aKG itself is able to recycle NO^•^ from peroxynitrite, but at normal pH. Hypoxia leads to acidic conditions and aKG is not able to counteract with peroxynitrite. A loss of NO^•^ in favor of ONOO^−^ seems to increase the cell survival of leukemic cells. As a consequence of oxidative metabolism O_2_^•−^ radical production is increased, e.g., by dysregulated iNOS and/or AKGDH, which can be easily converted by SOD to hydrogen peroxide and eliminated by CAT and/or GPx. Additional dysregulation of either SOD, CAT or GPx leaves more O_2_^•−^ to counteract NO^•^, leading to the formation of peroxynitrite which itself oxidizes thiols and nitrate proteins on tyrosine residues. We suggest that a balanced combination of 5-HMF and aKG are potential antioxidants in lowering peroxynitrite formation directly by an enzymatic upregulation of SOD, CAT and GPx with 5-HMF and a reduction in peroxynitrite to nitrite and/or NO^•^ in the presence of aKG. The aKG/5-HMF combination is able to inhibit nitration of tyrosine proteins synergistically, which leads to an increase in caspase activity and apoptosis in Jurkat cells in a concentration-dependent manner. This results into a decrease in cell growth in Jurkat cells as summarized in Figure 8 and Box 1.

Box 1Schematic description of dysregulation of NO^•^, antioxidative acting enzymes, reduced caspase activated cell death in leukemia cells in absence or presence of aKG and/or 5-HMF.Conditions in leukemic cells(1) Under hypoxic conditions leukemic cancer generates
RONS, such as ONOO^−^, whereas NO^•^ radicals are removed
by the reaction with O_2_*^−^ to generate additional ONOO^−^
(NO^•^/ONOO^−^ < 1); (2) alpha-ketoglutarate (mitochondrial or cytosolic) is removed by peroxynitrite under loss of molecular oxygen; (3) RONS react with proteins or amino acids forming increased CP or NP modulating
proteins, such as the p53 and p21 pathways or SOD, CAT, GPx, NOS and HIFα, but also carbohydrates, nucleic acids and fatty acids; (4) HIF-alpha increase because of the loss of alpha-ketoglutarate or oxygen; (5) SOD, CAT and GPx is modified and or inactivated by RONS; (6) cell growth is initiated over HIFα; (7) apoptosis is reduced and Cell death decrease.
Alpha-ketoglutarate added to leukemic cells:
1: Addition of alpha-ketoglutarate interferes with peroxides such as peroxynitrite; (2) recycling NO and accumulation of succinate; (3) succinate effects positively on HIFα, low cell growth, higher apoptosis and cell death; (4) reduction in RONS, such as ONOO^-^ or O_2_^•−^, implicate no modified proteins, amino acids, such as the p53 and p21 pathways, or SOD, CAT, GPx, NOS and HIFα, but also carbohydrates, nucleic acids and fatty acids and implicate again step 3.
5-HMF added to
leukemic cells:
1: Addition of 5-HMF
implicate saturation of molecular oxygen effecting positively HIFα. 2: Reduction in RONS, such as ONOO or O_2_^•−^ implicate no modified proteins, amino acids, such as the p53 and p21 pathways or SOD, CAT, GPx, NOS and HIFα,
but also carbohydrates, nucleic acids and fatty acids and implicate again step 3.

## 4. Materials and Methods

### 4.1. Materials

Media: RPMI 1640, FBS, DMEM, antibiotic–antimycotic solution (Thermofisher, Vienna, Austria); reagents: Cytofix–Cytoperm permeabilization Kit (Thermofisher, Vienna, Austria), FITC Active Caspase-3 Apoptosis Kit (BD Biosciences Kit; Allschwil, Germany), and WST-1 Cell Proliferation Reagent (Abcam; Cambridge, UK); chemicals: alpha-ketoglutarate, hydrochloric acid, mangan oxide and sodium nitrite (Sigma Aldrich, Vienna, Austria), and 5-hydroxymethyl-furfurale (5-HMF, Evonik Operation, Darmstadt, Germany); flasks (Falco^®^ Cell Culture; Corning Incorporated, New York, NY 14831, USA) were used.

### 4.2. Preparation of ONOO^−^

ONOO^−^ was prepared according to Hughes and Nicklin [27] by incipient mixing of equal volumes of 0.7 M H_2_O_2_ solution in 0.6 M HCl and 0.6 M NaNO_2_ on ice, followed immediately by termination of the reaction with 1.5 M NaOH. Surplus H_2_O_2_ was removed by the addition of a pinch of MnO_2_ and subsequent filtration of the suspension. The ONOO^–^ concentration was determined spectrophotometrically at 302 nm with an extinction coefficient of 1670 M^−1^ cm^−1^. Aliquots were stored at −70 °C until measurements.

### 4.3. ONOO^−^ Luminol Measurements Using aKG, 5-HMF and a Combination of aKG + 5-HMF

The consumption of peroxynitrite was measured via a luminescence technique with slight modifications [28]. A volume of 5 µL of 30 mM ONOO^−^ was transferred to a white 96-well microtitration plate (Nunc, Roskilde, Denmark). Different concentrations of aKG, 5-HMF and their combination were pipetted directly into the ONOO^−^. Immediately, after one second, luminol (400 µM 3-amino-phtalhydrazide in 10 mM PBS, pH 7.4) was added (total reaction volume 200 µL), and the chemiluminescence signal was detected on a BMG Lumistar plate reader (ServoLAB, Graz, Austria) for one signal. The luminescence signal was expressed in counts per second (cps).

### 4.4. Nitration of BSA with ONOO^−^

Nitration of tyrosine residues on BSA was established according to Greilberger et al. [4]. A quantity of 2 mg albumin was dissolved in 800 µL of 10 mM phosphate-buffered saline (pH 7.4) in the presence or absence of 0–36 mM aKG, 5-HMF and their combination. Three volumes (400 µL) of ONOO^−^ solution were added in three steps (every 10 min) to obtain a total volume of 2 mL with an ONOO^−^ end concentration of 12 mM. The total reaction time was 30 min at 37 °C in a closed tube. After the reaction, a 1 mL aliquot was used immediately for the determination of nitro-tyrosine-BSA via an ELISA technique. One milliliter of nitrated tyrosine BSA was dialyzed without any antioxidant in 10 mM PBS pH 7.4 overnight, changing the buffer solution 3 times (3 × 1 L). Volumes of 200 µL of several diluted BSA solutions (i.e., 3.12, 1.56, 0.78, and 0.39 µg/mL) in the absence or presence of αKG, 5-HMF and their combination (i.e., aKG + 5-HMF) were applied to a transparent 96-well microtitration plate (Nunc, Roskilde, Denmark) and incubated at 37 °C for 2 h. After three washes (washing buffer: 10 mM PBS + 0.01% Tween 20, pH 7.4) with an Auto Plate Washer EL X 450 (BioTek, Bad-Friedrichshall, Germany), 250 µL of blocking solution (0.2% I-Block in 10 mM PNS, pH 7.4) was pipetted into the wells. After 30 min at room temperature (RT), the plate was washed again three times with washing buffer. A volume of 200 µL of 1:1000 diluted rabbit anti-nitrotyrosine IgG in 10 mM PBS, pH 7.4, was applied and incubated for 1 h at 4 °C. After a further washing step (3000× *g*), 200 µL of goat anti-rabbit IgG-HRP (1:1000) was added to the samples and incubated for 1 hat RT. After at least six washes, a 200 µL solution was applied, and the reaction was stopped after 4 min at RT by adding a 100 µL stop solution. Using a Power WaveX plate photometer (Bio-Tek, Winooski, VT, USA), extinctions of standards and samples were measured at 490 nm.

### 4.5. Cell Culture

The Jurkat-J6 cell line (acute T-leukemic cells; Pune India) was received from the National Center for Cell Sciences. Resuspension was conducted in RPMI-1640 (containing glutamine, 10% FBS and 1% antibiotic–antimycotic solution) For further experiments, cells were incubated with a density of approximately 1,000,000 cells per mL.

### 4.6. Cell Proliferation Experiments

Cell proliferation experiments were performed using different aKG (0–3.5 µM), 5-HMF (0–4 µM) or aKG + 5-HMF concentrations (0, 1.7 µM aKG + 0.7 µM 5-HMF and 3.5 µM aKG + 1.7 µM 5-HMF) in medium for 24, 48 and 72 h with the cells as described earlier [4]. Cell growth was estimated with the CASY^®^ Cell Counter (Hoffmann-La Roche Ltd., Basel, Switzerland). Aliquots were used for further experiments.

### 4.7. Cytotoxicity Assay

A cytotoxicity assay (WST assay) was carried out to determine the viability of the Jurkat and HF-SAR cell lines as described [4]. Cells (200,000/mL) were applied onto transparent 24-well plates and incubated for 0, 24, 48 and 72 h in the absence or presence of aKG (0–3.5 µM), 5-HMF (0–4 µM) or aKG + 5-HMF (0, 1.7 µM aKG + 0.7 µM 5-HMF and 3.5 µM aKG + 1.7 µM 5-HMF) concentrations. After twice washing using DPBS 1X, fresh medium containing 10% WST-1 reagent was applied for a further 2 h. The absorbance was measured using the extinction of 450 nm in a spectrophotometric reader (Spectra Max Pro 384; Molecular Devices; San Jose, CA 95134, USA).

### 4.8. Caspase-3 Activity Measurements

Caspase-3-activated apoptosis was carried out as described in a recently published paper [4]. After incubation for 72 h in the absence or in the presence of 1.7 or 3.5 µM of aKG, 2 or 4 µM 5-HMF, 1.7 µg/mL aKG + 0.7 µM 5-HMF and 3.5 µM aKG + 1.3 µM 5-HMF, cells were separated by centrifugation at 3500× *g*. After several washing steps using cold PBS, 500 µL Cytofix–Cytoperm was added to the cells on ice at −20 °C. Antibody–FITC was used after incubation and the washing step was examination of the caspase-3 activity at 495/519 nm, respectively.

### 4.9. Statistical Analysis

*t*-Tests and ANOVA tests were used for comparing groups using SPSS 25 (SPSS Inc., Chicago, IL, USA). All values are given as the mean values and standard deviations. Statistical significance was considered to be at *p* < 0.05, with high significance at *p* < 0.01.

## 5. Conclusions

The combination of aKG and 5-HMF definitely reduces oxidative modifications, which are a hallmark in carcinogenesis and tumoral growth, including downregulation of antioxidative substances and antioxidative regulating enzymes of the Krebs cycle in the mitochondria but also in the cytosol. Both substances synergistically increase the antioxidative capacity, leading to a caspase-activated process, which eliminates Jurkat cells. The usage of this combination in solutions was effective in other clinical trials [10,12,13] without any increase in side effects; therefore, we suggest that the application of an IV solution containing both substances does have a potential role as a supporting therapy for leukemic patients in addition to the standard therapies used. Further clinical studies are therefore needed.

## Figures and Tables

**Figure 1 ijms-23-09034-f001:**
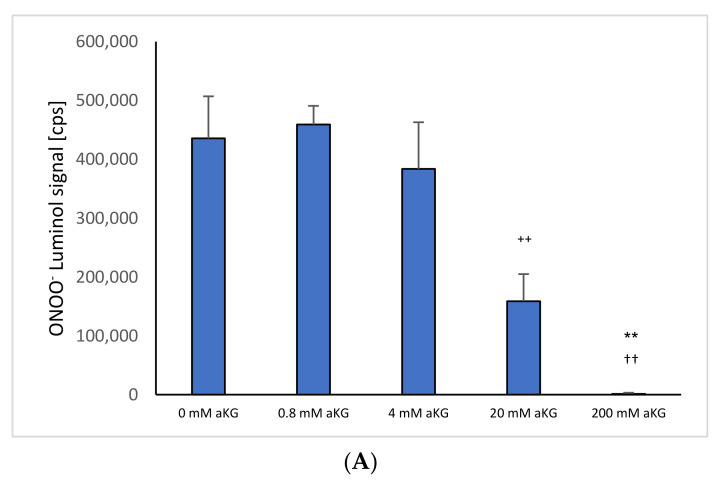
ONOO^−^ induced luminol measurements after 1 s in the absence or presence of (**A**) 0–200 mM aKG (*n* = 5); (**B**) 0–200 mM 5-HMF (*n* = 5); (**C**) aKG (0–200 mM) + 5-HMF (0–200 mM; *n* = 5). ** *p* < 0.01: significance between 20 and 200 mM aKG compared to 4 mM aKG; ^††^ *p* < 0.01: significance between 20 and 200 mM aKG; °° *p* < 0.01: significance between 0 mM 5-HMF and 20 and 200 mM 5-HMF; ^++^ *p* < 0.01: significance between 4 mM of combined aKG + 5-HMF and 20 mM of combined aKG + 5-HMF. ^§§^ *p* < 0.01: significance between 200 mM combined aKG + 5-HMF and 20 mM combined aKG + 5-HMF.

**Figure 2 ijms-23-09034-f002:**
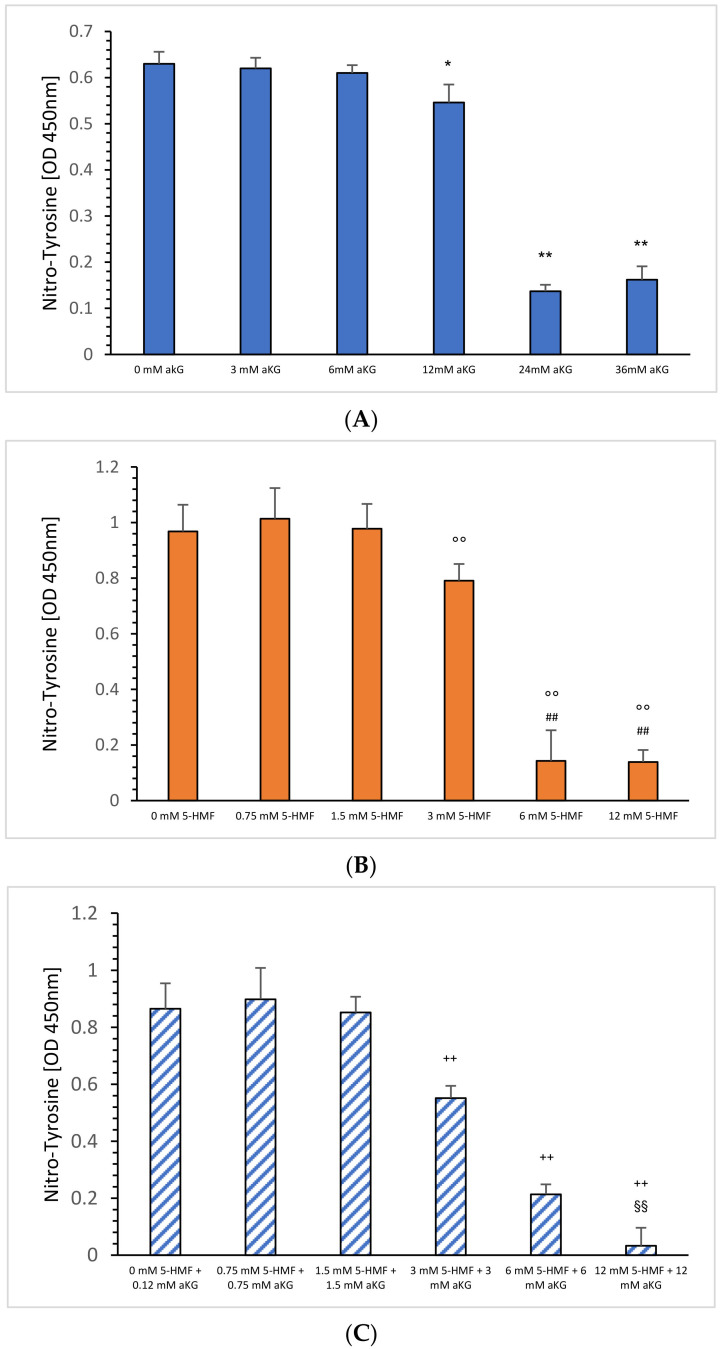
Estimation of nitro-tyrosine residues using ONOO^−^ as the nitrating reagent incubated with (**B**) 0.75, 1.5, 3, 6 and 24 mM 5-HMF (*n* = 5); 0.75, 1.5, 3, 6 and 24 mM combined solution of aKG + 5-HMF (**C**); (*n* = 5). (**A**) presents data from Greilberger et al. (2021) [4] on the antinitrating effect on tyrosine residues using 3, 6, 12, 24 and 36 mM aKG °° *p* < 0.01: significance between 3, 6 and 12 mM 5-HMF compared to 1.5 mM 5-HMF. ^##^ *p* < 0.01: significance between 6 and 12 mM 5-HMF compared to 3 mM 5-HMF; ^++^ *p* < 0.01: significance between combined 3, 6 and 12 mM 5-HMF + aKG and 1.5 mM combined 5-HMF + aKG; ^§§^ *p* < 0.01: significance between combined 6 mM 5-HMF + aKG and combined 12 mM 5-HMF + aKG; * *p* < 0.05: significance between 12 and 6 mM aKG; ** *p* < 0.01: significance between 24 or 36 mM and 12 mM aKG.

**Figure 3 ijms-23-09034-f003:**
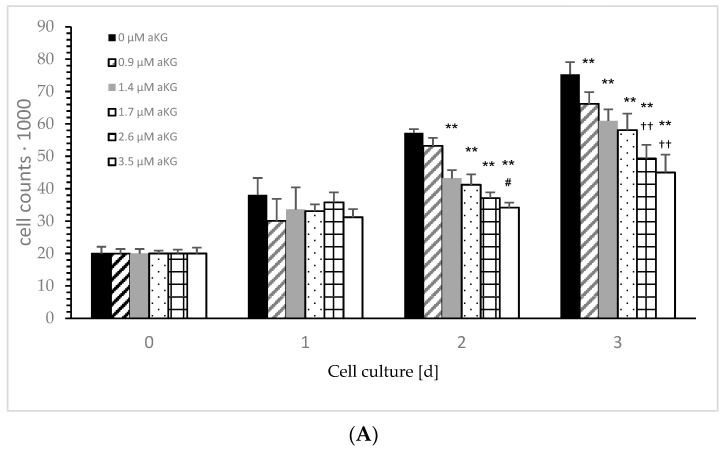
Cell growth of the Jurkat cell line in the absence or presence of different concentrations of 0, 0.9, 1.4, 1.7, 2.6 and 3.5 µM aKG (**A**) or 0, 1, 1.6, 2, 3 and 4 µM 5-HMF (**B**) after 24, 48 and 72 h of cultivation (*n* = 5). ** *p* < 0.01: significance between 0.9 µM aKG and 1.4 or 1.7 µM aKG; ^††^ *p* < 0.01: significance between 1.4 µM aKG and 2.6 or 3.5 µM aKG; ^#^ *p* < 0.05: significance between 0 and 1 µM 5-HMF; ^##^ *p* < 0.01: significance of 0 and 1.6 or 1 µM 5-HMF; ^&&^ *p* < 0.01; significance between 1 and 3 µM 5-HMF; ^^^ *p* < 0.05: significance between 3 and 4 µM 5-HMF; ^§§^ *p* < 0.01: significance between 0 µg/mL 5-HMF and 1.6 or 2 µM 5-HMF; ^$$^ *p* < 0.01: significance between 2 µg/mL 5-HMF and 3 or 4 µM 5-HMF.

**Figure 4 ijms-23-09034-f004:**
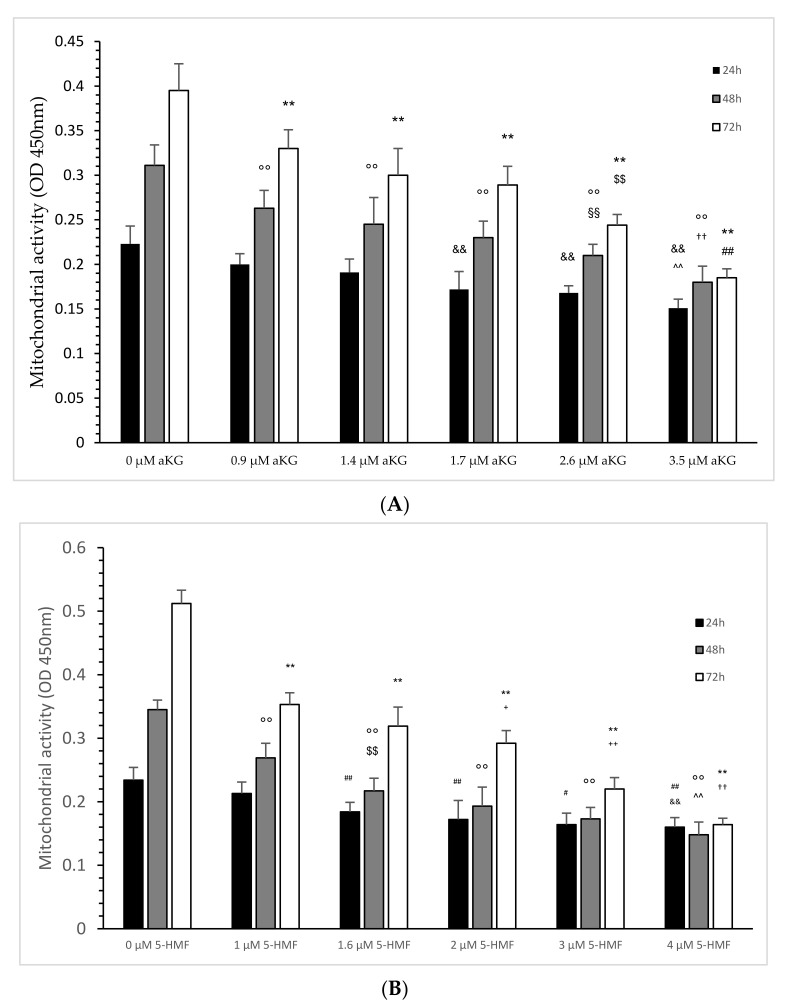
Mitochondrial activity of the Jurkat cell line in the absence or presence of different concentrations of aKG (**A**) or 5-HMF (**B**) estimated in absorbance after 24, 48 and 72 h of cultivation (*n* = 5). (**A**) On *24 h incubation:* ** *p* < 0.01: significance between 0 µM aKG and 0.9, 1.4, 1.7, 2.6 and 3.5 µM aKG; ^$$^ *p* < 0.01: significance between 2.6 and 1.7 µM aKG; ^##^ *p* < 0.01: significance between 2.6 and 1.7 µM aKG; *48 h incubation:* °° *p* < 0.01: significance between 0 µM aKG and 0.9, 1.4, 1.7, 2.6 and 3.5 µM aKG; ^§§^ *p* < 0.01: significance between 0.9 and 2.6 µM aKG; ^††^ *p* < 0.01: significance between 1.7 and 3.5 µM aKG; *72 h incubation:* ^&&^ *p* < 0.01: significance between 0 µM aKG and 1.7, 2.6 and 3.5 µM aKG; ^^^^ *p* < 0.01: significance between 1.4 and 3.5 µM aKG. (**B**) On *24 h incubation:*
^##^ *p* < 0.01: significance between 0 µM 5-HMF and 1 and 2 µM 5-HMF; ^#^ *p* < 0.05: significance between 1 and 3 µM 5-HMF; ^&&^ *p* < 0.01: significance between 1 and 4 µM 5-HMF. On *48 h incubation:* °° *p* < 0.01: significance between 0 µM 5-HMF and 1, 1.6, 2, 3 and 4 µM 5-HMF; ^$$^ *p* < 0.01: significance between 1 µg/mL 5-HMF and 1.6, 2 and 3 µM 5-HMF; ^^^^ *p* < 0.01: significance between 2 µg/mL and 4 µM 5-HMF. On *72 h incubation:* ** *p* < 0.01: significance between 0 µM 5-HMF and 1, 1.6, 2, 3 and 4 µM 5-HMF; ^+^ *p* < 0.05: significance between 1 and 2 µM 5-HMF; ^++^ *p* < 0.01: significance between 2 and 3 µM 5-HMF; ^††^ *p* < 0.01: significance between 3 and 4 µM 5-HMF.

**Figure 5 ijms-23-09034-f005:**
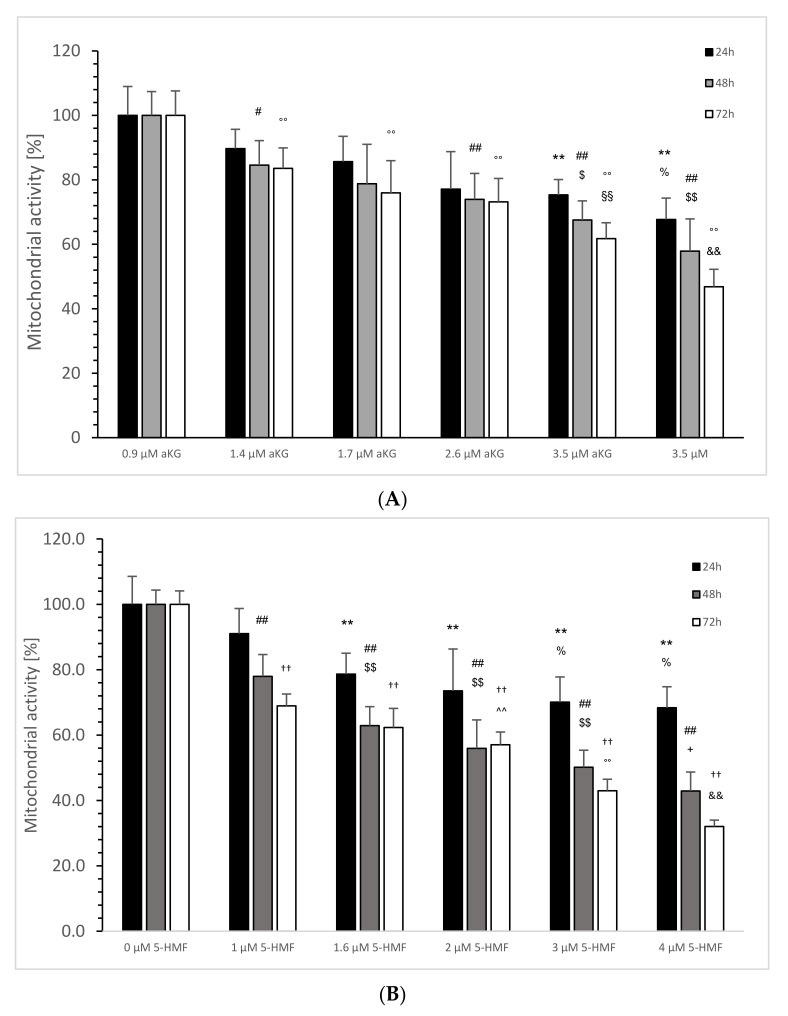
Calculated mitochondrial activity in percentage of the Jurkat cell line in the absence or presence of 0, 0.9, 1.4, 1.7, 2.6 and 3.5 µM aKG (**A**) or 0, 1, 1.6, 2, 3 and 4 µM 5-HMF (**B**) after 24, 48 and 72 h of cultivation (*n* = 5). (**A**) *On 24 h incubation:* ** *p* < 0.01: significance between 0 µM aKG and 2.6 and 3.5 µM aKG; ^%^ *p* < 0.05: significance between 2.6 and 3.5 µM; ^#^ *p* < 0.05: significance between 0 and 1.7 µM aKG; *48 h incubation:* ^##^ *p* < 0.01: significance between 0 µM aKG and 2.6 and 3.5 µM aKG; ^$^ *p* < 0.01: significance between 0.9 and 2.6 µM aKG; ^$$^ *p* < 0.01: significance between 0.9 and 3.5 µM aKG; *72 h incubation* °° *p* < 0.01: significance between 0 µM aKG and 0.9, 1.4, 1.7, 2.6 and 3.5 µM aKG; ^§§^ *p* < 0.01: significance between 0.9 and 2.6 µM aKG; ^&&^ *p* < 0.01: significance between 2.6 and 3.5 µM aKG; (**B**) *24 h incubation:* ** *p* < 0.01: significance between 0 µM 5-HMF and 1, 1.6, 2, 3 and 4 µM 5-HMF; ^%^ *p* < 0.05: significance between 1 µM 5-HMF and 3 and 4 µM 5-HMF; *48 h incubation:* ^##^ *p* < 0.01: significance between 0 µM 5-HMF and 1, 1.6, 2, 3 and 4 µM 5-HMF; ^§§^ *p* < 0.01: significance between 1 µM 5-HMF and 1.6, 2 and 3 µM 5-HMF; ^+^ *p* < 0.05: significance between 2 and 4 µM 5-HMF; *72 h incubation:* ^††^ *p* < 0.01: significance between 0 µM 5-HMF and 1, 1.6, 2, 3 and 4 µM 5-HMF; ^^^^
*p* < 0.01: significance between 1 and 2 µM 5-HMF; °° *p* < 0.01: significance between 2 and 3 µM 5-HMF; ^&&^ *p* < 0.01: significance between 3 and 4 µM 5-HMF.

**Figure 6 ijms-23-09034-f006:**
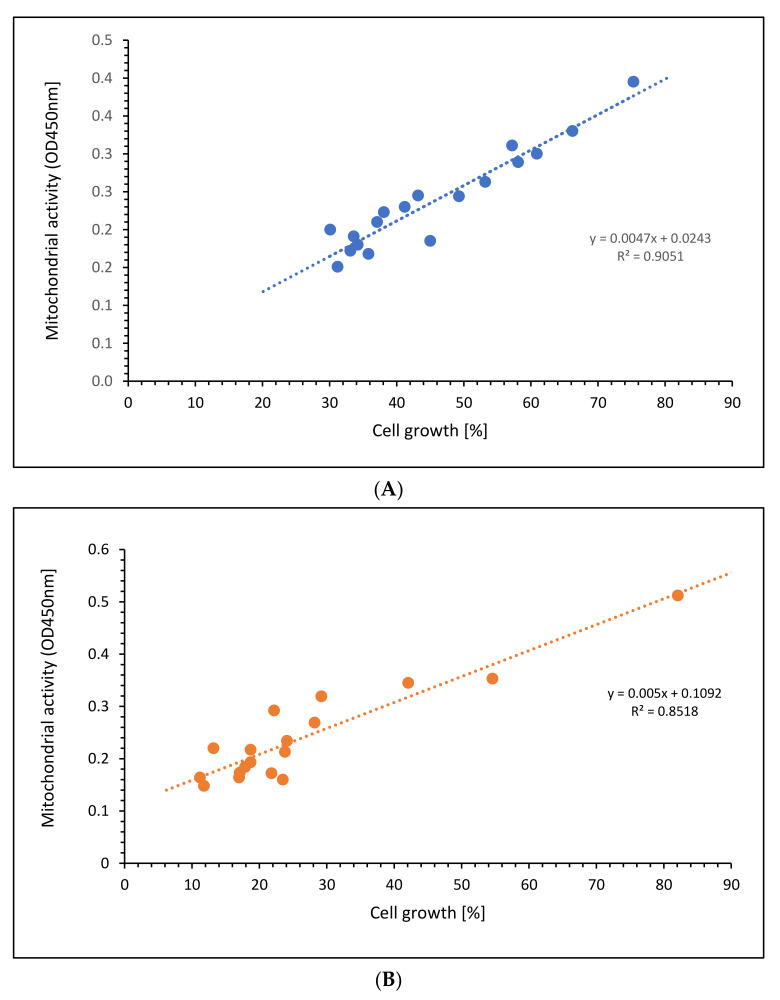
Regression of cell growth against mitochondrial activity using different concentrations of (**A**) aKG or (**B**) 5-HMF on Jurkat cell lines.

**Figure 7 ijms-23-09034-f007:**
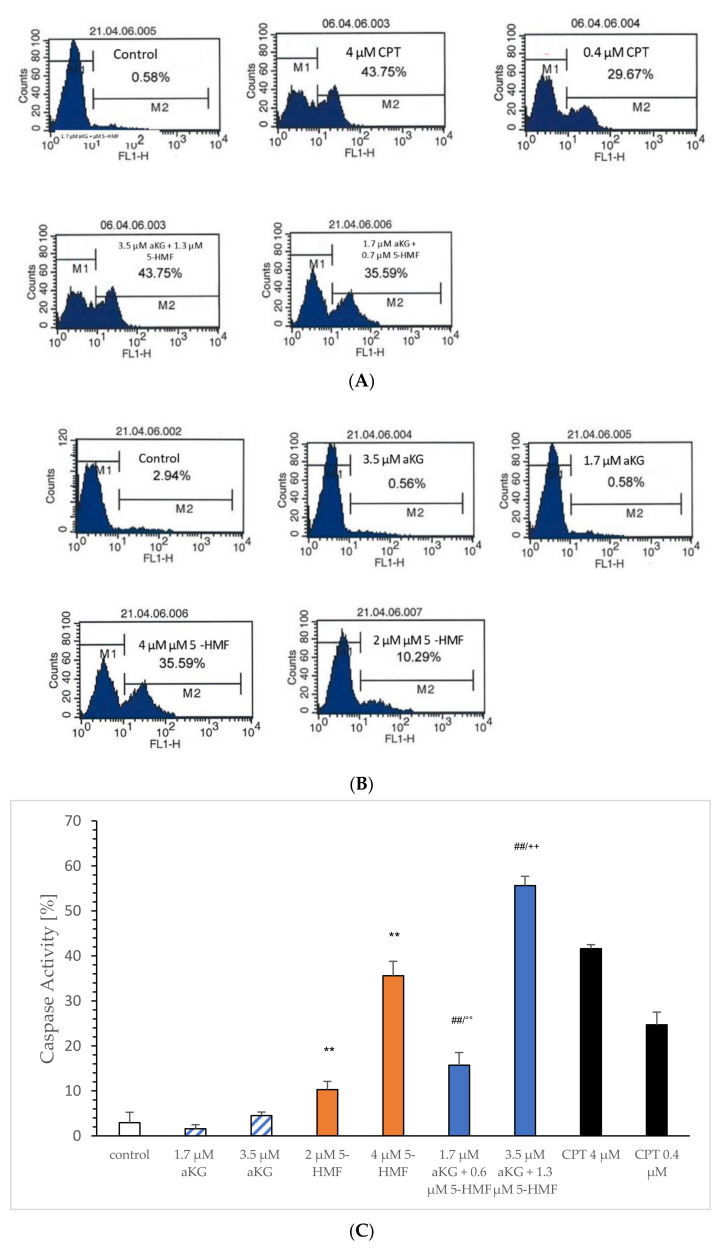
Representative original immunohistoblots of the caspase-activated apoptosis of the Jurkat cell lines using (**A**) 0.4 and 4 µM CPT, 1.7 µM aKG + 0.6 µM 5-HMF and 3.4 µM aKG + 1.3 µM 5-HMF; (**B**) 0.4 and 4 µM CPT, 1.7 and 3.4 µM aKG, 2 and 4 µM 5-HMF, caspase activity of 0.4 and 4 µM CPT, 1.7 and 3.4 µM aKG, 2 and 4 µM 5-HMF (*n* = 3); (**C**) correlation of caspase-activated apoptosis against different concentrations of aKG, 5-HMF, CPT and combination of aKG + 5-HMF on Jurkat cell lines (**D**). ** *p* < 0.01: significance between control and 2 or 4 µg/mL 5-HMF; ^##^ *p* < 0.01: significance between the control and 1.7 µM aKG + 0.7 µM 5-HMF and 3.5 µM aKG + 1.3 µM 5-HMF; °° *p* < 0.01: significance between 0.7 µM 5-HMF and 3.5 µM aKG and 2 µM 5-HMF; ^++^ *p* < 0.01: significance between 3.5 µM aKG + 1.3 µM 5-HMF and 4 µM 5-HMF.

**Figure 8 ijms-23-09034-f008:**
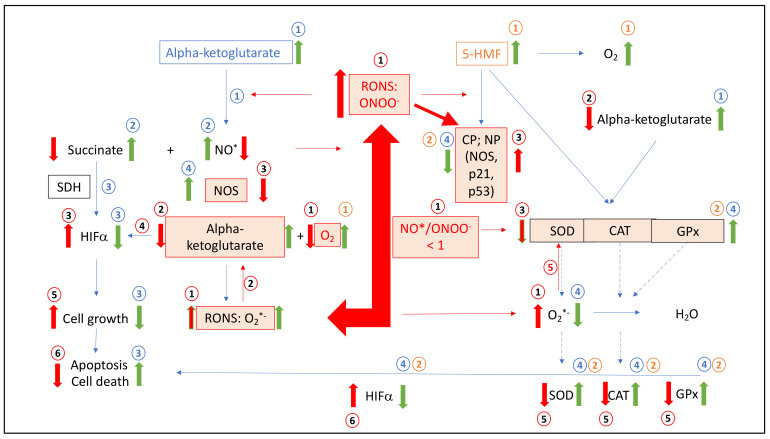
Summary Figure (i) The situation of leukemic cells in the case of RONS and the ratio of NO/ONOO, modified proteins (i.e., HIFα, SDH, SOD, CAT, GPx and NOS), carbohydrates, nucleic acids, alpha-ketoglutarate and cell growth; (ii) Mode of action of alpha-ketoglutarate and 5-HMF added ingredients to leukemic cells to increase apoptosis and cell death via multiple antioxidative enzymatic and nonenzymatic pathways. RONS: reactive oxygen and nitrogen species; O_2_^•−^: superoxide anion radical; NO^•^: nitrosyl radical; ONOO^−^: peroxynitrite; HIFα: hypoxia-inducing factor alpha; NOS: nitrosyl-synthase; SOD: superoxide dismutase; CAT: catalase; GPx: glutathione peroxidase; CP: carbonyl proteins; NP: nitrated proteins; SDH: succinate dehydrogenase; alpha-ketoglutarate: in leukemic cells (mitochondrial and/or cytosolic). Conditions in leukemic cells (Box 1).

## Data Availability

The data presented in this study are available in article.

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
