# Peer review of "Alpha-Ketoglutarate or 5-HMF: Single Compounds Effectively Eliminate Leukemia Cells via Caspase-3 Apoptosis and Antioxidative Pathways"

_ijms, 2022, doi:10.3390/ijms23169034_

Round 1

Reviewer 1 Report

The present manuscript, entitled, Alpha-Ketoglutarate or 5-HMF: Single compounds eliminate effectively leukemia cells via caspase-3, apoptosis and antioxidative pathways, would be a nice contribution in the field of leukemia research. However, the manuscript needs some modifications prior to publication.

The manuscript requires native English check and editing to improve the language and correct English. For example, line 22, 5-HMF effects, line 24, Luminol infduced, line 27, Aliquots of Jurkat cells. What is the meaning of aliquots of Jurkat cells?

Line 56-57, please mention some keywords that represent the study.

The introduction part requires more information about the background of the study, models used, hypotheses, etc.

Figures could be improved.

Page 10, there is a figure without figure legends. It seems that the figure is repeated again on page 11 as Figure 4. Please check.

Please include a concrete conclusion based on the study results.

Please include a graphical abstract or a summary figure if possible.

Author Response

Rev 1:

Dear reviewer, thank you for your comments. We have definitely improved the manuscript after using your comments. We took also a native speaker for better understanding of our presented results and hypotheses.

With best regards

Greilberger

Reviewer 2 Report

The authors report alpha-ketoglutarate or 5-HMF.

1.      There were no control data in the Figure 2A, 2B and 2C. Please clarify these things.

2.      The concentration of aKG and 5-HMF were different in this study (e.g., mM or μg/ml). The authors should be consistent with the concentration.

3.      The mechanistic analysis was not performed in this study. The authors should provide the representative immunohistochemical and immunoblot data.

4.      In Figure 7, the authors should provide the representative quadrant flow cytometry data.

5.      In Figure 8, the authors should provide the immunoblot data.

6.      The authors should add the graphical summary data in a Figure. It will be benefit for the reader.

Author Response

(The authors gave the same response as above.)

Round 2

Reviewer 2 Report

none

This manuscript is a resubmission of an earlier submission. The following is a list of the peer review reports and author responses from that submission.